# Novel ABCD1 and MTHFSD Variants in Taiwanese Bipolar Disorder: A Genetic Association Study

**DOI:** 10.3390/medicina61030486

**Published:** 2025-03-11

**Authors:** Yi-Guang Wang, Chih-Chung Huang, Ta-Chuan Yeh, Wan-Ting Chen, Wei-Chou Chang, Ajeet B. Singh, Chin-Bin Yeh, Yi-Jen Hung, Kuo-Sheng Hung, Hsin-An Chang

**Affiliations:** 1Department of Psychiatry, Tri-Service General Hospital, School of Medicine, National Defense Medical Center, Taipei 11490, Taiwan; light-0100@outlook.com (Y.-G.W.); 400010428@mail.ndmctsgh.edu.tw (C.-C.H.); fantine7520@ndmctsgh.edu.tw (T.-C.Y.); friend1584@gapps.ndmctsgh.edu.tw (W.-T.C.); chinbinyeh@gmail.com (C.-B.Y.); 2Department of Radiology, Tri-Service General Hospital, School of Medicine, National Defense Medical Center, Taipei 11490, Taiwan; weichou.chang@gmail.com; 3The Institute for Mental and Physical Health and Clinical Translation (IMPACT), School of Medicine, Barwon Health, Deakin University, Geelong 3220, Australia; a.singh@deakin.edu.au; 4Division of Endocrinology and Metabolism, Department of Internal Medicine, Tri-Service General Hospital, School of Medicine, National Defense Medical Center, Taipei 11490, Taiwan; metahung@yahoo.com; 5Center for Precision Medicine and Genomics, Tri-Service General Hospital, School of Medicine, National Defense Medical Center, Taipei 11490, Taiwan

**Keywords:** bipolar disorder, disorder, genome-wide association study (GWAS), genetic association study, genetic variants, association study, Taiwanese Han population, population, ABCD1 gene, MTHFSD gene, fatty acid metabolism, linkage disequilibrium (LD), X-linked inheritance, population-specific variants, minor allele frequency (MAF), Taiwan precision medicine initiative (TPMI)

## Abstract

*Background and Objectives:* In recent years, bipolar disorder (BD), a multifaceted mood disorder marked by severe episodic mood fluctuations, has been shown to have an impact on disability-adjusted life years (DALYs). The increasing prevalence of BD highlights the need for better diagnostic tools, particularly those involving genetic insights. Genetic association studies can play a crucial role in identifying variations linked to BD, shedding light on its genetic underpinnings and potential therapeutic targets. This study aimed to identify novel genetic variants associated with BD in the Taiwanese Han population and to elucidate their potential roles in disease pathogenesis. *Materials and Methods:* Genotyping was conducted using the Taiwan Precision Medicine Array (TPM Array) on 128 BD patients and 26,122 control subjects. Following quality control, 280,177 single nucleotide polymorphisms (SNPs) were analyzed via chi-square tests, and linkage disequilibrium (LD) analyses were employed to examine the associations among key SNPs. *Results:* Eleven SNPs reached significance (*p* < 10^−5^), with the variant rs11156606 in the ABCD1 gene—implicated in fatty acid metabolism—emerging as a prominent finding. LD analysis revealed that rs11156606 is strongly linked with rs73640819, located in the 3′ untranslated region, suggesting a regulatory role in gene expression. Additionally, rs3829533 in the MTHFSD gene was found to be in strong LD with the missense variants rs3751800 and rs3751801, indicating potential alterations in protein function. *Conclusion:* These findings enhance the genetic understanding of BD within a Taiwanese cohort by identifying novel risk-associated variants and support the potential for using these markers in early diagnosis and targeted therapeutic strategies.

## 1. Introduction

Bipolar disorder (BD) is a clinically severe mood disorder with a lifetime prevalence of 4% [1]. From 1990 to 2017, the incidence of BD increased by 47.74%, and the disability-adjusted life years (DALYs) associated with the condition increased by 54.4% during this period [2], taking away years of healthy functioning from individuals with the illness.

Currently, BDs are primarily diagnosed by careful assessment of behavior combined with subjective reports of abnormal experiences to group patients into disease categories standardized in The Diagnostic and Statistical Manual of Mental Disorders (DSM) and The International Statistical Classification of Diseases and Related Health Problems (ICD). However, BD types I and II are difficult to diagnose accurately in clinical practice, particularly in early stages, in which phenomenology can be extremely similar. When a true bipolar patient is assessed in a depressive phase, there is a relative risk of more than 40% of being mistakenly diagnosed as MDD [3,4]. Moreover, nearly a quarter of adults (22.5%) and adolescents with major depressive disorder (MDD), followed up for a mean length of 12–18 years, developed BD [5]. Underestimated prevalence and difficulties in diagnosis have resulted in an overarching need for more precise and early diagnostic methods.

The first GWAS was published in 2005. Since then, researchers have become optimistic about the prospect of the genome–disease association approach [6]. This is particularly important in psychiatry, given the unconvincing and inconsistent evidence from candidate gene studies and the genetic architecture for most diseases that seem to be polygenic [7]. Since the first genome–disease association study of BD in 2007, a handful of risk loci have been identified, notably *ANK3* [8], *NCAN* [9], *CACNA1C* [10], and *ODZ4* [11]. In recent years, an increasing number of studies have reported hundreds of genes and proteins related to BD, many of which have been suggested as potential biomarkers for this disease, including *BDNF* [12], *RELN* [13], and *ANK* [8]. More importantly, BD exhibits a polygenic architecture, in which numerous genes can collectively influence its pathogenesis through diverse mechanisms—such as gene regulation, epigenetic modifications, and gene–environment interactions—where the cumulative burden of many small-effect variants contributes to individual risk [7].

Findings from genome-wide association studies (GWAS) in Western populations often have limited transferability to Taiwanese and other non-Western groups due to differences in genetic architecture, allelic frequency distributions, linkage disequilibrium patterns, and distinct environmental and lifestyle factors [14]. In this study, we aimed to demonstrate the capabilities of genome–disease association analysis using the TPM array to detect genetic variations of bipolar disorder (BD) in the Taiwanese population. This analysis aids in predicting specific Taiwanese risk-relevant single nucleotide polymorphisms (SNPs), which could be applied in the diagnosis of BD, by investigating individuals who participated in the Tri-Service General Hospital (TSGH) genetic study project. Additionally, we focused on identifying the potential mechanisms of the most influential characterized variants, as well as related variants in ABCD1 and MTHFSD, which may further inform treatment strategies for BD.

## 2. Materials and Methods

### 2.1. Study Participants and Ethical Approval

Our study is part of the TSGH clinical genetics project under the Taiwan Precision Medicine Initiative (TPMI), spearheaded by Academia Sinica in collaboration with 16 medical centers nationwide. The TPMI has established a comprehensive biobank and research infrastructure, enabling the collection of DNA samples for genetic analyses and facilitating both retrospective and prospective access to participants’ disease phenotypes for longitudinal monitoring. A distinctive feature of this program is its reciprocal approach, providing participants with their genetic risk profiles and opportunities to join validation studies aimed at refining disease risk-prediction models and implementing risk-stratified healthcare protocols.

We recruited patients diagnosed with BD (ICD-10 code F31) based on diagnostic interviews conducted by clinical psychiatrists according to the DSM, Fifth Edition (DSM-5) criteria. Genotyping data were collected from the TPM array, the third-generation Taiwan Biobank SNP array designed by Academia Sinica and Thermo Fisher Scientific specifically for the Taiwan Precision Medicine Initiative (TPMI) project. This array facilitates large-scale genetic studies within the Taiwanese population by capturing genetic variation unique to the Taiwan Han Chinese. It comprises genome-wide SNPs, markers for complex diseases, pharmacogenomic indicators, and variants exclusive to Taiwanese genetics. This emphasis on Taiwanese diversity ensures relevance to East Asian populations while highlighting rare variations not commonly observed elsewhere [15].

Participants in this study were recruited from medical centers and genotyped by Academia Sinica. DNA extraction and SNP identification were performed as follows: First, each participant’s genomic DNA was extracted and purified from 3 mL of peripheral blood collected in EDTA vacutainers using the QIAsymphony SP system (QIAGEN, Hilden, Germany). Second, the purified DNA from each participant was loaded onto the TPM array chip, and genome-type signals were detected using the Axiom GeneTitan system (Thermo Fisher Scientific, Sunnyvale, CA, USA). Conversion and quality control of SNP calling and sample annotation were performed using the Axiom Analysis Suite (Thermo Fisher Scientific, Sunnyvale, CA, USA).

### 2.2. Disease Association Analysis

Figure 1 illustrates the steps involved in the disease association analysis. Initially, 128 individuals diagnosed with BD (ICD-10 code F31), determined through diagnostic interviews conducted by clinical psychiatrists according to the DSM, Fifth Edition (DSM-5) criteria, were recruited from the TSGH. The control group comprised 26,122 individuals without BD. Comprehensive participant information is presented in Table 1.

To analyze the genotyping data from the TPM array, we first excluded SNPs with a typing call rate below 80%. Next, we removed variants with a minor allele frequency under 0.05 or a Hardy–Weinberg equilibrium *p*-value below 1 × 10^−6^. We then performed a chi-squared test for association, setting the significance level to *p* < 0.05, using PLINK 1.9 software (https://zzz.bwh.harvard.edu/plink/ (v1.90, accessed on 7 October 2024)) [16]. To address potential inflation of test statistics, we implemented Genomic Control (GC), developed by Devlin and Roeder [17], as our primary *p*-value adjustment method.

### 2.3. Variant Annotations and Functional Analysis

Genes were identified for variant annotations utilizing the RefSeq Database (https://www.ncbi.nlm.nih.gov/refseq/ (accessed on 5 December 2024)), as described in wANNOVAR (https://wannovar.wglab.org/ (accessed on 5 December 2024)) [18]. To compare the allele frequency in different racial populations, the public domain databases 1000 Genomes [19], Genome Aggregation Database (gnomAD) [20], and Taiwan BioBank (https://taiwanview.twbiobank.org.tw/index (accessed on 20 January 2025)) [15] were used. Further characterization of the genes associated with the identified variants involved examining their biological functions and molecular pathways. The biological processes of the genes, as defined by Gene Ontology [21], were elucidated using Enrichr (https://maayanlab.cloud/Enrichr/ (accessed on 20 January 2025)) [22].

### 2.4. Linkage Disequilibrium (LD) Analysis

LD analysis helps researchers understand which genes are likely to be functionally relevant to a trait or disease, as well as the number and location of the contributing genes [23]. In our study, we uploaded the variants *rs11156606* and *rs3829533* obtained from the association study using the LDproxy Tool, which is part of LDlink (https://ldlink.nih.gov/?tab=home (accessed on 14 December 2024)) [24,25]. Analysis parameters were as follows: 1. Genome Build Version: GRCH37 (*rs11156606*) and GRCH38 (*rs3829533*). 2. Population: CDX (Chinese Dai in Xishuangbanna, China), CHB (Han Chinese in Beijing, China), and CHS (Southern Han Chinese). 3. LD measurements: D′ (D prime) and R^2^ (R-squared) were calculated based on allele frequencies. The measured values of D′ and R^2^ range from 0 to 1, where a value of 1 indicates complete disequilibrium, and a value of 0 indicates complete equilibrium. The base-pair window is 500,000 bp. Regulatory potential prediction was based on FORGEdb [26].

## 3. Results

### 3.1. Demographics of the Selected Patients

Table 1 lists the demographic composition of the participants enrolled in this study stratified by sex and age group. Within the BD cohort, female participants slightly outnumbered male participants, with 69 females and 59 males. The control group was more extensive, comprising 11,793 males and 14,329 females, indicating a demographic balance that mirrors the expected population distributions. The age distribution within the BD cohort revealed the highest prevalence in the 50–59 age demographic, with a progressive decrease observed in both the younger and older cohorts. By contrast, the control group demonstrated substantial inclusivity across all adult age ranges, ensuring adequate age-matched controls for robust comparative analyses.

### 3.2. Study Workflow

Figure 1 shows the gene–disease association study pipeline implemented in our investigation to identify genetic susceptibilities associated with BD. This study differentiated between a cohort case of 128 individuals diagnosed with BD and a sizable control cohort consisting of 26,122 participants. Genotyping was performed using the Taiwan Precision Medicine (TPM) array chip, designed to accurately detect a comprehensive spectrum of SNPs. The initial genotypic yield from this high-throughput platform was an extensive array of 493,852 SNPs.

The subsequent stage involved a filtering process to ensure data integrity and relevance, in which a substantial number of SNPs were excluded from the initial pool owing to factors such as low minor allele frequency, deviations from Hardy–Weinberg equilibrium, or missing data. After filtering out the data that could potentially confound the analysis, a curated set of 280,177 high-quality SNPs was obtained.

The selected SNPs were analyzed using the chi-square test to detect significant associations between SNP frequencies and BD incidence. Variants exhibiting a *p*-value of less than 10^−5^ were deemed significant and flagged for in-depth investigation to ascertain their potential as risk factor indicators.

### 3.3. Genetic Variants Identified, Statistical Analysis, and Significance

According to our association study results, there were 11 genetic variants that demonstrated significant (*p* < 10^−5^) associations with the condition (refer to the data points above the red line in the Manhattan plot in Figure 2). The Q-Q plot (Figure 2) shows significant deviations above the diagonal line, suggesting a potential link with BD. The genomic inflation factor, lambda, derived from the median of the observed and expected chi-square values, was close to 1. This indicates minimal inflation and suggests that our test statistics are robust. Therefore, we were able to confidently interpret the results, affirming the minimal bias in the findings of our study.

The variants identified with high significance (specifically rs6427761, rs6658422, rs7530898, rs74419649, rs79221549, rs6496485, rs3829533, rs755981, rs6081464, rs7289613, and rs11156606) were distributed across various chromosomes. These SNPs were associated with several coding and non-coding genes, such as ABCD1, C20orf78, DPP10, LOC284930, LPP, MTHFSD, NR5A2, and non-coding RNA genes, including LINC01221 and NTRK3-AS1 (Table 2). Each variant demonstrated a high odds ratio (>1.5), indicating a robust association with BD.

We conducted a search of the GWAS Catalog (https://www.ebi.ac.uk/gwas/ (accessed on 20 January 2025)) to further understand the novelty and relevance of these findings. Our search revealed that all variants except rs11156606 have not been previously reported to be associated with BD or any physical disease. Notably, rs11156606, located at the 153,741,041 bp position on the Chromosome X, was recorded in the GWAS Catalog (accession ID GCST000821) as being associated with both BD and SCZ [27]. In our study, this variant showed a particularly high odds ratio of 2.36, suggesting a more than two-fold increase in the risk of BD in carriers of the allele.

The allele frequencies of the identified variants in our study population varied significantly, ranging from 15.7% to 33.2% in cases and 7.3% to 21.6% in controls. These variations highlight the distinct prevalence of genetic markers. Moreover, the allele frequencies in our study were closely aligned with those recorded in the Taiwan Biobank, which profiles the genetics of healthy Taiwanese individuals. This similarity suggests that the genetic makeup in our control group was representative of the general healthy population in Taiwan (Table 3).

Furthermore, a comparison with other global populations revealed that the allele frequencies in our control group were similar to those found in Asian populations. Notably, the frequencies of rs6496485, rs3829533, rs755987, and rs6081464 were higher in our study than in African, American, and European populations (Table 3). This suggests that these four variants may serve as specific biomarkers for BD in the Taiwanese population, offering the potential for more targeted diagnostic approaches.

### 3.4. Functional Annotations of Risk Genes

As shown in Table 4**,** we identified 64 significant Gene Ontology (GO) biological process terms with *p*-values less than 0.05. These terms are predominantly related to biological functions such as fatty acid metabolism, protein localization, potassium ion transport, vascular transport, and viral replication. Notably, fatty acid metabolism emerged as a major significant GO term, particularly associated with the variant rs11156606 located in the ABCD1 gene.

Analysis of similar GO terms revealed three key cluster networks: (1) Fatty Acid Metabolism/Oxidative Stress Regulation; (2) Lipid Homeostasis and Membrane Potential Regulation; and (3) Localization of Lipid Transport and Membrane Proteins, as shown in Figure 3. Additionally, four ABCD1-related GO terms were identified with direct relevance to neuronal functions: GO:0106027 (neuron projection organization), GO:1990535 (neuron projection maintenance), GO:0043217 (myelin maintenance), and GO:0042552 (myelination). These findings suggest that mutations in ABCD1 play a crucial role in the functioning of the nervous system, potentially contributing to the pathology of BD and other neurological conditions.

### 3.5. LD Analysis

The significant variant rs11156606, identified in the association study and located in intron 7 of ABCD1, was linked through LDproxy analysis (Figure 4) to a rare missense mutation, rs782720024 (c.436T>A, p.Phe146Ile), in exon 1. This mutation exhibits low allele frequencies, approximately 0.002, specifically within the EAS and Taiwanese populations, indicating its rarity in the Chinese population. This suggests that individuals in populations with this variant may have a higher risk of BD. Additionally, another SNP, rs73640819 (c.131G>A), found in the 3′ UTR with high LD values, plays a crucial role in gene expression regulation, including transcription, translation, RNA stability, and localization. Its allele frequency was notably the lowest in the Taiwanese population at 0.07 (Appendix A), highlighting population-specific genetic differences and their potential impact on disease susceptibility and gene regulation mechanisms.

Furthermore, LDproxy analysis revealed that the variant rs3829533, located in exon 6 of MTHFSD, is closely associated with two other SNPs, rs3751800 (D′ = 1, R^2^ = 0.987) and rs3751801 (D′ = 1, R^2^ = 1), based on data from the Chinese population. Notably, the associated SNPs located in MTHFSD exon 8 resulted in missense coding changes (rs3751800: c.727C>T/p. Arg243Cys and rs3751801: c.730G>C/p.Ala244Pro) (Appendix A and Figure 5). (Figure 5) This finding is significant because it suggests that while the Taiwanese population may possess the synonymous variant rs3829533 at a relatively high frequency (0.238 in cases and 0.138 in controls, as shown in Table 2), which does not alter coding, the associated missense variants rs3751800 and rs3751801 could potentially modify the molecular function of MTHFSD owing to changes in amino acids, thereby increasing the risk of BD. Appendix A illustrates that the allele frequencies of the two variants in the East Asian and Taiwanese populations (0.16 and 0.15, respectively) were higher than those in the African, American, and European populations.

Additionally, FORGEdb scores, as depicted in Figure 4 and Figure 5, predicted the likelihood of genetic variants functioning as regulatory elements. These scores range from 0 to 10 and are based on various regulatory DNA datasets, including transcription factor (TF) binding and chromatin accessibility. The FORGEdb scores for rs11156606, rs73640819, rs3829533, rs3751800, and rs3751801 are 7, 10, 7, 8, and 8, respectively. These high scores indicated a significant regulatory effect on ABCD1 and MTHFSD, further suggesting that these variants could be specific risk factors for BD in the Taiwanese population.

We investigated the nucleotide polymorphisms in the three variants within the MTHFSD gene (Figure 6). The variants rs3829533 and rs3751801 were synonymous; the former retained the amino acid threonine at position 175 of the MTHFSD protein translated from exon 6. Conversely, rs3751800 represented a missense variant, whereas its LD-associated variant rs3751801 induced amino acid substitutions at positions 243 and 244 of the MTHFSD protein (arginine to cysteine and alanine to proline, respectively). These changes occurred within exon 8 of MTHFSD.

## 4. Discussion

This study represents an advancement in our understanding of the genetic underpinnings of BD, particularly in the Taiwanese population. The identified 11 novel genetic variants associated with BD could contribute to earlier and more accurate diagnosis by helping clinicians identify individuals with a heightened genetic risk. If these variants are confirmed through larger replication studies and integrated into a polygenic risk score (PRS), clinicians might better distinguish BD from unipolar depression, especially during ambiguous early phases. This stratification could lead to more targeted monitoring of at-risk individuals, potentially reducing the risk of misdiagnosis and facilitating timely intervention.

Furthermore, the alignment of allele frequencies with those in the Taiwan Biobank reinforces the relevance of our findings to Taiwanese demographics, providing a foundation for future studies to explore genetic risk factors and their implications in a targeted manner. BD is a complex disease with unknown causes, involving various factors such as demographics, genetics, and environment, some of which have strong evidence supporting their link to BD. Therefore, this study not only enriches our genetic understanding of BD but also underscores the critical need for tailored genetic research in diverse populations to uncover the nuanced nature of complex psychiatric disorders.

For the analysis, sometimes a Bonferroni-corrected threshold is applied to account for the large number of tests (e.g., *p* < ~1.8 × 10^−7^ for 280,177 SNPs in our study). However, due to our modest sample size and the exploratory nature of this work, we adopted *p* < 1 × 10^−5^ as a suggestive significance cutoff. Conventional methods like Bonferroni and FDR, though effective at controlling for multiple comparisons, do not address GWAS-specific confounders such as linkage disequilibrium (LD) and population stratification. In particular, the Bonferroni method assumes complete independence among tests, which is rarely the case in GWAS because of extensive LD, leading to overly conservative thresholds that can mask true associations. To mitigate these issues, we implemented Genomic Control (GC) as our primary *p*-value adjustment method [17]. GC directly accounts for inflation in test statistics due to population structure by calculating and applying a genomic inflation factor (λ). This approach corrects for ancestry-related allele frequency differences, a critical concern in GWAS of genetically diverse cohorts. A comparative analysis of Bonferroni, FDR, and GC is provided in Revised Appendix A, demonstrating that GC maintains robust control of type I error while better accommodating population stratification [28,29].

There is growing attention on the interaction between specific genes and the environment as well as the suggested involvement of fatty acid metabolism and neuronal function in the disorder [30]. The variant rs11156606, located intronically in the *ABCD1* gene, presents intriguing possibilities regarding its mechanistic role in BD. The protein encoded by this gene is a member of the ATP-binding cassette (ABC) transporter superfamily. ABC genes are divided into seven distinct subfamilies (ABC1, MDR/TAP, MRP, ALD, OABP, GCN20, and White). This protein is a member of the ALD subfamily, which is involved in the peroxisomal import of fatty acids and/or fatty acyl-CoAs in organelles [31]. It also plays a crucial role in cellular processes such as membrane transport [32,33,34], circulatory system stability [35], lipid transport, and homeostasis [35]. Dysregulation of ABC transporters has been implicated in various diseases, including Mendelian disorders [32], cancer, and drug resistance [34]. Several studies have indicated that fatty acid metabolism plays a significant role in BD development and manifestation. In this study, Leclercq S. et al. tested PUFA as potent inducers of the ABCD genes. The expression levels of *ABCD2* and *ABCD3* were significantly higher in n-3-deficient rats than in rats fed ALA- or DHA-supplemented diets, indicating sensitivity towards dietary PUFA [36]. In a study comparing patients with BD and healthy controls, subjects with BD had distinctly lower levels of omega-3 eicosapentaenoic acid (EPA) and higher omega-6 arachidonic acid levels, coupled with increased plasma IL-6 and TNF-α levels [37]. We established a connection between the genotype of the fatty acid desaturase gene cluster and altered polyunsaturated fatty acid levels in BD. In one study, fatty acids in the erythrocyte membranes of patients with bipolar manic disorder and healthy controls were analyzed using thin-layer chromatography and gas chromatography. The results showed lower levels of arachidonic acid (20:4n-6) and docosahexaenoic acid (22:6n-3) in BD patients with BD compared to normal controls [38]. The association of the *ABCD1* variant *rs11156606* with altered fatty acid metabolism in BD underscores its potential role in the pathophysiology of the condition. This link, highlighted by the distinct PUFA profiles in patients with BD, suggests that dysregulation of ABC transporters, which are key in lipid transport and homeostasis, could be a critical factor in BD development.

The discovery of *rs11156606* within intron region 7 of *ABCD1* via disease–gene association interpretation exemplifies the effectiveness of such research in identifying genetic variants that are potentially crucial for disease predisposition. This study combined data from people with SCZ and BD to find shared genetic factors and revealed that SNP *rs11156606* is associated with BD and SCZ [27]. Notably, the unique presence of the rare allele *rs782720024* in East Asian and Taiwanese populations, along with the discovery of *rs73640819*, which shows high LD and is situated at the 3′ UTR of *ABCD1*, underlines the significance of population-specific genetic studies. These findings reveal the intricate nature of gene regulation and its influence on disease, highlighting how allele frequency variations across populations, especially their scarcity in Taiwanese individuals, may shed light on the unique regulatory mechanisms that affect gene expression and disease outcomes.

The discovery of the high LD-associated variant *rs73640819* in the 3′ untranslated region (3′ UTR) of *ABCD1* highlights its importance beyond a mere sequence tailing the coding region. The 3′-UTR plays a pivotal role in the regulation of gene expression by influencing mRNA stability, translation efficiency, and subcellular localization, thereby fine-tuning protein synthesis. This region is integral to the complex post-transcriptional gene regulation machinery and acts as a critical regulator of the pathway from DNA transcription to mRNA translation into proteins [39]. Such insights open avenues for further exploration of how variations in the 3′-UTR affect gene function and disease mechanisms, indicating a rich field for ongoing research.

The variant with c.131G>A changes introduces polyadenylation in the 3′-UTR, highlighting the nuanced role of 3′-UTRs in genetic regulation. Unlike coding regions, 3′-UTRs are marked by AT-rich sequences that play a critical role in directing polyadenylation to these regions, thus influencing mRNA stability and cellular localization. This highlights the importance of 3′-UTRs in controlling gene expression through management of the mRNA lifecycle. Additionally, the diversity of mRNA isoforms attributed to alternative cleavage and polyadenylation (APA) allows for the production of multiple 3′ UTR variants from a single gene, each with unique regulatory capabilities [40]. This complexity underscores the significance of APA in the broader context of post-transcriptional regulation, opening paths for further investigation of its role in gene expression diversity and implications for cellular function and disease.

The Methenyltetrahydrofolate Synthetase-Domain-Containing (*MTHFSD*) gene plays a pivotal role in diverse physiological processes, including vascular health and cancer susceptibility, through its cytoplasmic activity in RNA binding, which facilitates crucial cellular functions [41]. It encodes a protein that is essential for the conversion of 5,10-methylenetetrahydrofolate to 5-methyltetrahydrofolate, a key step in the re-methylation of homocysteine to methionine, which is vital for cell health. Notably, the potential of S-adenosyl methionine (SAM) to alter lipid raft arrangements and influence receptor and transporter functions has been linked to therapeutic prospects in depression, BD, and SCZ [42]. Recent studies, including those by Lyu, highlight nutrient deficiencies, particularly in the conversion process from homocysteine to methionine, as is common in BD and SCZ [43], echoing Ozdogan et al.’s findings from measuring homocysteine levels in patients [44]. Animal models such as those of Akahoshi et al. suggest that a high-methionine diet could induce BD-like behaviors [45], pointing to a potential connection between high methionine intake and emotional states. Further research on *MTHFSD* expression identified it as a top differentially expressed gene in SCZ at various illness stages [46]. A 2016 study on methamphetamine-associated psychosis (MAP) identified *MTHFSD* as a candidate biomarker for RNA degradation [47], highlighting its role in folate catabolism and methionine metabolism. This connection is further supported by a meta-analysis by Hsieh et al., which found that individuals with BD generally have lower serum folate levels than healthy controls [48], underscoring the enzyme’s role in maintaining active folate levels for vital biosynthetic pathways. LD analysis revealed two correlated variants (*rs3751800*: c.727C>T/p. Arg243Cys and *rs3751801*: c.730G>C/p.Ala244Pro) within the *MTHFSD* gene, indicating its potential as a disease marker for BD in the Taiwanese population owing to its effect on *MTHFSD* function. These findings emphasize *MTHFSD*’s critical involvement in essential biological processes and their impact on human health, warranting further exploration of its mechanisms and therapeutic potential.

A limitation of our study is the small sample size, which restricts statistical power and the robustness of our findings, particularly affecting the analysis of high-OR SNPs. Future studies with larger cohorts are needed to better validate our results and enhance the predictive accuracy of the genetic markers. The small sample size also affects the statistical power to thoroughly validate the rs11156606 variant and its potential X-linked inheritance pattern, where recruitment of additional patients and their family members is required to further investigate rs11156606 and elucidate its functional implications in disease pathogenesis. In a meta-analysis of individuals of East Asian (EAS) ancestry, two BD-associated loci—rs117130410 (*p* = 3.68 × 10^−8^, OR = 1.31) and rs174576 (*p* = 7.78 × 10^−9^, OR = 0.86)—were identified [49]. These findings differ from our results, which may be attributed to differences in demographics, analytical methods, or threshold criteria. Future studies could aim to validate the variants identified in both studies. Another limitation of our study is the 80% call-rate threshold, which helps maintain reliable detection but excludes lower call-rate SNPs that might otherwise increase genetic diversity. We plan to enroll additional participants in a new clinical trial to increase sample size and use genetic imputation strategies, thereby enhancing both the robustness of our findings and the overall SNP diversity.

## 5. Conclusions

This study successfully identified novel genetic variants associated with BD in the Taiwanese Han population. These findings illuminate the intricate genetic basis of BD and highlight the significance of research on diverse populations. The discovered variants implicated pathways related to fatty acid metabolism, lipid homeostasis, and neuronal function, suggesting potential targets for future research. This study advances our understanding of BD’s complex genetic architecture, potentially paving the way for improved diagnostic and targeted therapeutic approaches.

## Figures and Tables

**Figure 1 medicina-61-00486-f001:**
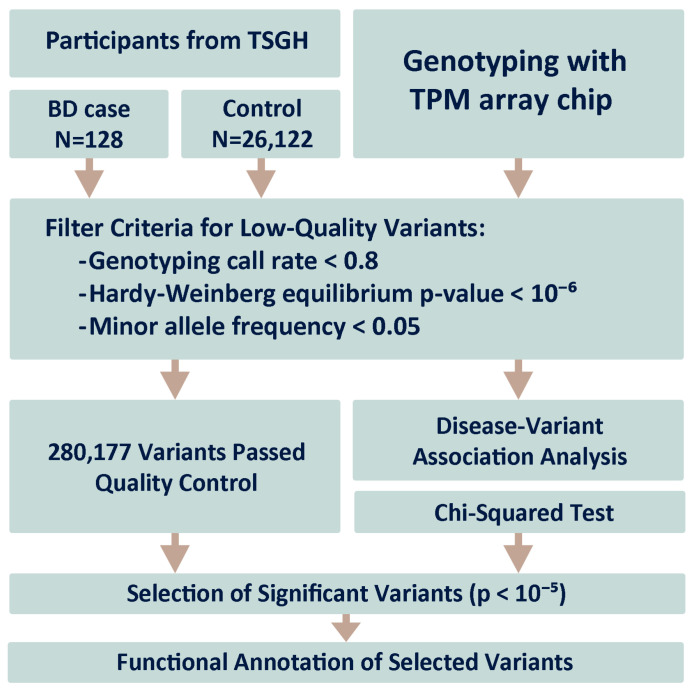
Disease–gene association analysis pipeline used in this study. The case group (128 bipolar disorder patients) and the control group (26,122 participants). Genotyping was performed using the Taiwan Precision Medicine (TPM) Array Chip. Data from 493,852 SNPs were filtered, and 280,177 SNPs were passed through the chi-squared test to detect risk factors.

**Figure 2 medicina-61-00486-f002:**
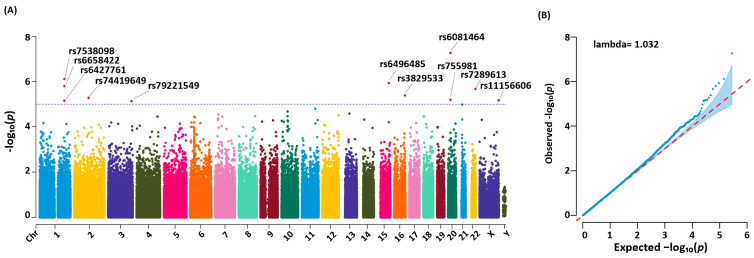
Manhattan and Q-Q plots of association study results in BD patients. (**A**) A total of 280,177 variants were detected in TSGH participants through a chi-squared test. Eleven highly significant variants were selected based on a *p*-value < 10^−5^ (red spots above the blue dashed line): rs7538098, rs6427761, rs7538098, rs74419649, rs79221549, rs6496485, rs3829533, rs75581, rs6081464, rs7289613, and rs11156606. (**B**) The Q-Q (quantile–quantile) plot shows the quality of the association test. The observed distribution of low *p*-values (with high observed −log_10_(*p*) values) indicates true associations with bipolar disorder. The genomic inflation factor (λ) was calculated using the median of the observed and expected chi-squared values.

**Figure 3 medicina-61-00486-f003:**
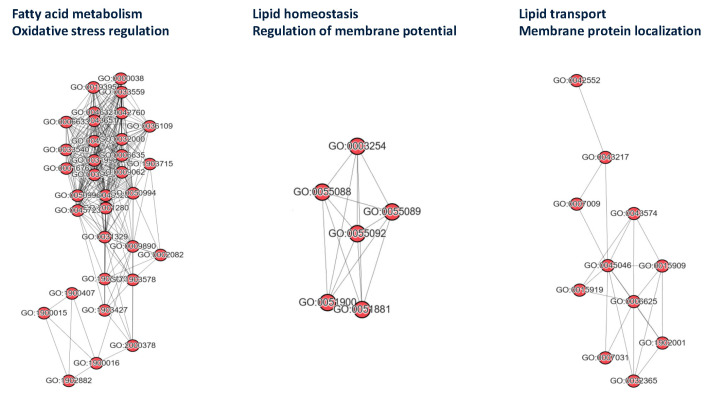
Significant GO BP term similarity networks of the ABCD1 analysis. The BP functions of the genes were identified using the GO database, with terms selected based on *p*-values < 0.05. The GO terms are listed in Table 4. Similarity networks were constructed using Navier–GO.

**Figure 4 medicina-61-00486-f004:**
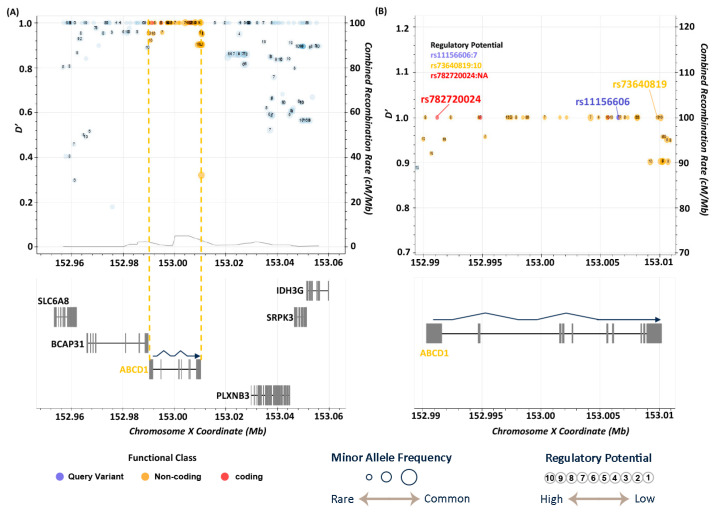
LD analysis of the variant rs11156606 on Chromosome X (152.96–153.06 Mb). (**A**) Overview of LD-associated variants within the ABCD1 gene region, highlighted between the orange dashed lines. The transcription direction is indicated by the arrow from 3′ to 5′ on Chromosome X. (**B**) Close-up of the genomic locations of rs11156606, rs782720024, and rs73640819. The missense variant rs782720024 is found in exon 1 of ABCD1, while rs73640819 is located in the non-coding 3’ UTR region.

**Figure 5 medicina-61-00486-f005:**
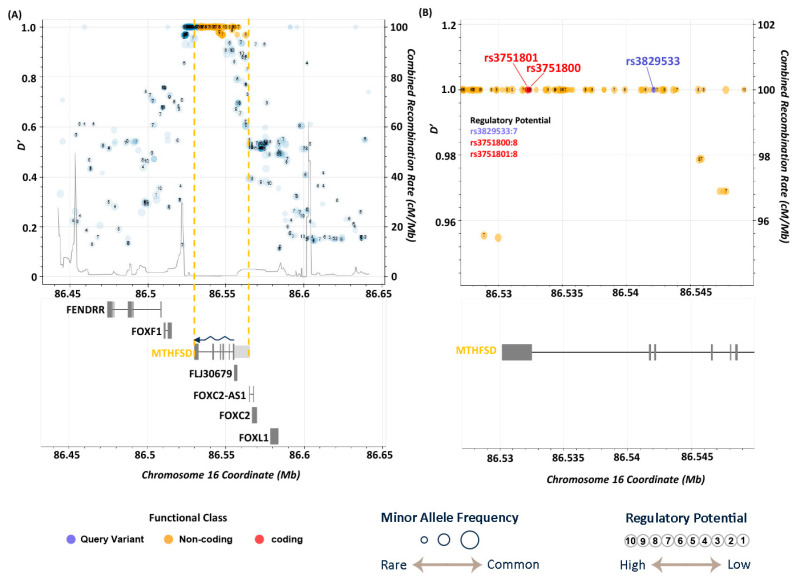
LD analysis of the variant rs3829533 on Chromosome 16 (86.45–86.65 Mb). (**A**) Overview of LD-associated variants within the MTHFSD gene region, including its promoter sequence (light gray box). The gene region is marked between the orange dashed lines, and the arrow indicates the transcription direction from 3′ to 5′ on Chromosome 16. (**B**) Close-up of the locations of rs3829533, rs3751800, and rs3751801. The missense variants rs3751800 and rs3751801 are located in exon 8 of MTHFSD.

**Figure 6 medicina-61-00486-f006:**
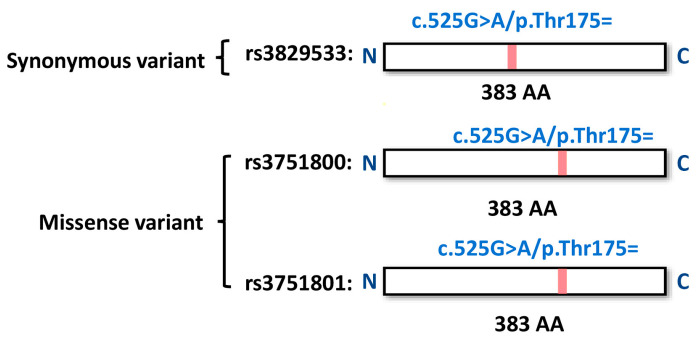
Coding changes in rs3829533, rs3751800, and rs3751801 in the MTHFSD gene. The synonymous variant rs3829533 preserves the amino acid threonine at position 175 of the MTHFSD protein (translated from exon 6). In contrast, the LD-associated variants rs3751800 and rs3751801 result in amino acid substitutions at positions 243 and 244, respectively: arginine to cysteine and alanine to proline. These alterations occur within exon 8 of MTHFSD.

**Table 1 medicina-61-00486-t001:** Participant Information in this study.

Group		BD	Control	*p*-Value
Sex	Male	59	11,793	0.63
	Female	69	14,329	
Age (Average ± SD)	>90	0	197 (93 ± 2)	-
	80–89	1 (81 ± 0)	895 (83 ± 3)	-
	70–79	7 (73 ± 3)	2970 (73 ± 3)	0.86
	60–69	25 (64 ± 3)	5681 (64 ± 3)	0.26
	50–59	28 (54 ± 3)	5124 (55 ± 3)	0.54
	40–49	25 (44 ± 3)	4096 (44 ± 3)	0.16
	30–39	24 (35 ± 3)	3424 (35 ± 3)	0.21
	20–29	15 (25 ± 3)	3002 (25 ± 3)	0.56
	10–19	3 (15 ± 2)	586 (15 ± 3)	0.07
	<9	0	147 (8 ± 1)	-

**Table 2 medicina-61-00486-t002:** Variants obtained from the association study results with high significance.

CHR	Position	SNP ID	Ref ^a^	Alt ^b^	*p*-Value ^c^	Odds Ratio	Region	Relative Gene
1	199219779	rs6427761	A	G	1.56 × 10^−6^	2.02	intergenic	LINC01221;NR5A2
1	199223240	rs6658422	C	T	7.01 × 10^−6^	1.96	intergenic	LINC01221;NR5A2
1	199262608	rs7538098	T	G	7.63 × 10^−7^	2.08	intergenic	LINC01221;NR5A2
2	114648288	rs74419649	T	C	5.17 × 10^−6^	1.88	intronic	DPP10
3	188587806	rs79221549	C	T	7.47 × 10^−6^	2.09	intronic	LPP
15	88454801	rs6496485	G	A	1.18 × 10^−6^	2.01	intergenic	NTRK3-AS1;MRPL46
16	86542131	rs3829533	C	T	4.06 × 10^−6^	1.95	exonic	MTHFSD
20	19027381	rs755981	G	T	6.48 × 10^−6^	1.81	intergenic	C20orf78;SLC24A3
20	19031000	rs6081464	T	C	5.41 × 10^−8^	2.09	intergenic	C20orf78;SLC24A3
22	480s87568	rs7289613	C	T	2.14 × 10^−6^	2.06	intergenic	LOC284930;MIR3201
X	153741041	rs11156606	A	C	6.69 × 10^−6^	2.36	intronic	ABCD1

^a^ Allele in the control. ^b^ Allele in the case. ^c^ Filter by *p* < 10^−5^.

**Table 3 medicina-61-00486-t003:** Allele frequencies of variants from other genetic projects.

This Study				TPMI ^a^	TWBank ^b^	1000 g ^c^				gnomAD ^d^				
SNP ID	Alt	Case	Control	-	-	AFR ^e^	AMR	EAS	EUR	AFR	AMR	EAS	FIN	NFE
rs6427761	G	0.231	0.129	0.134	0.134	0.200	0.098	0.160	0.022	0.195	0.087	0.123	0.044	0.025
rs6658422	T	0.215	0.122	0.127	0.127	0.290	0.100	0.150	0.047	0.249	0.101	0.115	0.089	0.043
rs7538098	G	0.224	0.122	0.127	0.126	0.290	0.099	0.150	0.041	0.268	0.086	0.118	0.076	0.044
rs74419649	C	0.273	0.167	0.168	0.157	0.011	0.260	0.180	0.110	0.021	0.318	0.160	0.213	0.133
rs79221549	T	0.169	0.089	0.089	0.084	0.002	0.037	0.091	0.033	0.006	0.039	0.084	0.038	0.028
rs6496485	A	0.242	0.137	0.137	0.131	0.076	0.085	0.140	0.076	0.091	0.076	0.138	0.074	0.084
rs3829533	T	0.238	0.138	0.138	0.150	0.008	0.110	0.160	0.110	0.023	0.106	0.144	0.102	0.092
rs755981	T	0.332	0.216	0.220	0.205	0.550	0.420	0.210	0.430	0.543	0.370	0.213	0.561	0.470
rs6081464	C	0.285	0.160	0.163	0.152	0.022	0.110	0.160	0.130	0.039	0.075	0.158	0.140	0.127
rs7289613	T	0.203	0.110	0.113	0.120	0.160	0.069	0.095	0.098	0.168	0.041	0.121	0.133	0.096
rs11156606	C	0.157	0.073	0.075	0.081	0.698	0.135	0.099	0.115	0.640	0.131	0.087	0.110	0.119

^a^ Taiwan Precision Medicine Initiative. ^b^ Taiwan Biobank. ^c^ 1000 Genomes Project (global minor allele frequency). ^d^ gnomAD (genome) allele frequencies. ^e^ AFR: African. AMR: American. EAS: East Asian. EUR: European. FIN: Finnish. NFE: Non-Finnish European.

**Table 4 medicina-61-00486-t004:** GO annotation of variant genes ^a^.

GO_ID	Term	*p*-Value	Gene
GO:0042758	long-chain fatty acid catabolic process	0.003	ABCD1
GO:0042760	very long-chain fatty acid catabolic process	0.004	ABCD1
GO:2001280	positive regulation of unsaturated fatty acid biosynthetic process	0.004	ABCD1
GO:0032000	positive regulation of fatty acid beta-oxidation	0.005	ABCD1
GO:0043574	peroxisomal transport	0.005	ABCD1
GO:1900407	regulation of cellular response to oxidative stress	0.005	ABCD1
GO:0043217	myelin maintenance	0.006	ABCD1
GO:1990535	neuron projection maintenance	0.006	ABCD1
GO:0046321	positive regulation of fatty acid oxidation	0.007	ABCD1
GO:0051900	regulation of mitochondrial depolarization	0.007	ABCD1
GO:0055089	fatty acid homeostasis	0.007	ABCD1
GO:1902882	regulation of response to oxidative stress	0.007	ABCD1
GO:0030497	fatty acid elongation	0.008	ABCD1
GO:0031998	regulation of fatty acid beta-oxidation	0.008	ABCD1
GO:0036109	alpha-linolenic acid metabolic process	0.008	ABCD1
GO:0045723	positive regulation of fatty acid biosynthetic process	0.008	ABCD1
GO:1903427	negative regulation of reactive oxygen species biosynthetic process	0.008	ABCD1
GO:0033540	fatty acid beta-oxidation using acyl-CoA oxidase	0.009	ABCD1
GO:1903715	regulation of aerobic respiration	0.009	ABCD1
GO:0002082	regulation of oxidative phosphorylation	0.010	ABCD1
GO:0009792	embryo development ending in birth or egg hatching	0.010	NR5A2
GO:0046320	regulation of fatty acid oxidation	0.010	ABCD1
GO:0106027	neuron projection organization	0.010	ABCD1
GO:0003254	regulation of membrane depolarization	0.011	ABCD1
GO:0009890	negative regulation of biosynthetic process	0.011	ABCD1
GO:1900016	negative regulation of cytokine production involved in inflammatory response	0.011	ABCD1
GO:1902001	fatty acid transmembrane transport	0.011	ABCD1
GO:0032365	intracellular lipid transport	0.013	ABCD1
GO:0043651	linoleic acid metabolic process	0.013	ABCD1
GO:0045046	protein import into peroxisome membrane	0.013	ABCD1
GO:0050994	regulation of lipid catabolic process	0.013	ABCD1
GO:0050996	positive regulation of lipid catabolic process	0.013	ABCD1
GO:2000378	negative regulation of reactive oxygen species metabolic process	0.014	ABCD1
GO:0007031	peroxisome organization	0.015	ABCD1
GO:0015919	peroxisomal membrane transport	0.016	ABCD1
GO:0006625	protein targeting to peroxisome	0.017	ABCD1
GO:0045070	positive regulation of viral genome replication	0.017	NR5A2
GO:1903426	regulation of reactive oxygen species biosynthetic process	0.017	ABCD1
GO:0043266	regulation of potassium ion transport	0.018	DPP10
GO:1904062	regulation of cation transmembrane transport	0.018	DPP10
GO:0015909	long-chain fatty acid transport	0.019	ABCD1
GO:0000038	very long-chain fatty acid metabolic process	0.020	ABCD1
GO:0051881	regulation of mitochondrial membrane potential	0.024	ABCD1
GO:1900015	regulation of cytokine production involved in inflammatory response	0.026	ABCD1
GO:0042552	myelination	0.028	ABCD1
GO:0006635	fatty acid beta-oxidation	0.031	ABCD1
GO:1903078	positive regulation of protein localization to plasma membrane	0.031	DPP10
GO:1904377	positive regulation of protein localization to cell periphery	0.031	DPP10
GO:0033559	unsaturated fatty acid metabolic process	0.032	ABCD1
GO:1901379	regulation of potassium ion transmembrane transport	0.032	DPP10
GO:1903578	regulation of ATP metabolic process	0.033	ABCD1
GO:0019395	fatty acid oxidation	0.035	ABCD1
GO:0048524	positive regulation of viral process	0.037	NR5A2
GO:0007009	plasma membrane organization	0.038	ABCD1
GO:0031329	regulation of cellular catabolic process	0.038	ABCD1
GO:0055088	lipid homeostasis	0.038	ABCD1
GO:0045069	regulation of viral genome replication	0.039	NR5A2
GO:0009062	fatty acid catabolic process	0.041	ABCD1
GO:0006633	fatty acid biosynthetic process	0.042	ABCD1
GO:0055092	sterol homeostasis	0.042	ABCD1
GO:1903076	regulation of protein localization to plasma membrane	0.047	DPP10
GO:0001676	long-chain fatty acid metabolic process	0.049	ABCD1
GO:0010232	vascular transport	0.049	SLC24A3
GO:0150104	transport across blood–brain barrier	0.050	SLC24A3

^a^ Selected based on *p*-value < 0.05.

## Data Availability

The data presented in this study are available upon request from the corresponding author due to legal regulations of the database.

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
