# Peer review of "Novel ABCD1 and MTHFSD Variants in Taiwanese Bipolar Disorder: A Genetic Association Study"

_medicina, 2025, doi:10.3390/medicina61030486_

Round 1

Reviewer 1 Report

Comments and Suggestions for Authors

This is a genome-wide analysis a sample of in 128 BD cases and 26122 controls from the Taiwan Precision Medicine Array (TPM Array), aiming to identify significant single nucleotide polymorphisms (SNPs) associated with BD. The authors identified the variant rs11156606 in the ABCD1 gene, which plays a role in fatty acid metabolism a process potentially connected to BD pathophysiology. Subsequent linkage disequilibrium (LD)analysis of rs11156606 revealed a strongly associated variant, rs73640819, located within the ABCD1  3’ untranslated region (3’UTR), suggesting a regulatory role on ABCD1 RNA functionality. Additionally, the variant rs3829533 was in strong LD with rs3751800 and 41 rs3751801, which significantly affected the coding sequence of the MTHFSD gene.

The main limitation of the paper lies in its very limited sample size of BD cases. This could impact on the likelihood of type 1 error and determine false positive.

I have several points:

  • Characteristics of the TPM array should be provided
  • There is lack of information on clinical assessment of BD patients (structured interviews? Diagnosis according to which criteria?assessment tools?
  • Why the sample size is so limited
  • The presence of a limited sample size of cases motivates the presentation of PPV NPV and sensitivity specificity
  • Could the authors compare their results with other GWAS in Asian ancestry (see last PGC BD GWAS (O’Donnel et al. Nature)

These points are critical to permit publication of the paper.

Author Response

Thank you very much for your thoughtful review. We sincerely appreciate your time and insights. Please refer to the attached PDF for further details and clarifications. 

Reviewer 2 Report

Comments and Suggestions for Authors

1) Why is there such a large number of people taken for the control group?

2) The control group includes cohorts of 90+ years, 80-90, as well as 10-19 and less than 9 years. Why were these data taken, especially for the study group? Was the sample checked for outliers and extreme points? If so, how? If there were outliers and extreme points, how were they included in the analysis? If excluded, how did this affect the quality of the final data and subsequent results?
3) When genotyping on a chip (TPMarray), based on what data was the genotyping call rate threshold of 80% selected, below which potential genetic variants were cut off?
4) From 493_852 SNPs, 280_177 were selected. Next, when conducting the chi-square analysis, variants with a significance level below 0.00001 (10 to the minus 5 power) were selected. However, when applying the Bonferroni correction for multiple comparisons, the significance level is adjusted to 0.0000002 (10 to the minus 7 power) (calculated as a standard significance level of 0.05 divided by the number of tests (SNPs), and 280_177 were selected). On what basis was this cutoff threshold chosen? If any other correction for multiple comparisons was used, this should be described in the text and it should be explained why this method was chosen.
5) When analyzing the rs11156606 variant, located on the X chromosome and, therefore, linked inheritance - was this taken into account in the further analysis? If so, how was it taken into account? If not, why not?
6) Line 222 - How might the annotated biological function viral replication be related to the disease under study bipolar disorder?
7) Linkage Disequilibrium Analysis - Why were D' and R2 measures chosen for the analysis rather than, for example, Bayesian method or Haplotype Block Analysis?
8) Describe your study limitations.

Author Response

(The authors gave the same response as above.)

Reviewer 3 Report

Comments and Suggestions for Authors

The manuscript presents a genetic association study on bipolar disorder in the Taiwanese population, identifying novel single nucleotide polymorphisms associated with BD. The study is well-structured and methodologically sound, utilizing data from the Taiwan Precision Medicine Array and employing linkage disequilibrium analysis. However, some aspects require clarification, additional context, or methodological improvements. 
1. The introduction could elaborate on why Taiwanese-specific genetic studies are needed. Are previous GWAS findings from Western populations not fully applicable?
2. While BD diagnostic difficulties are mentioned, the introduction should cite specific statistics on misdiagnosis rates or overlap with major depressive disorder.
3. Given the large control group (n=26,122), the low number of BD cases (n=128) might reduce statistical power. Was a power analysis performed to determine sample adequacy?
4. Were any covariates (age, sex, psychiatric comorbidities, medication history, environmental factors) controlled for in the analysis? This is crucial in genetic studies.
5. Given the polygenic nature of BD, a polygenic risk score analysis could strengthen the predictive power.
6. Odds ratios are presented, but confidence intervals should be included to assess statistical precision.
7. Was a Bonferroni correction or FDR adjustment applied to control for multiple comparisons?
8. The link between ABCD1, fatty acid metabolism, and BD is theoretically plausible, but the discussion lacks citations of experimental evidence.
9. How can these findings improve BD diagnosis or treatment? Would genetic screening for these SNPs be clinically useful?

Author Response

(The authors gave the same response as above.)

Reviewer 4 Report

Comments and Suggestions for Authors

thank you very much.

I am happy to review your manuscript.

I hope the recommendations  can help you to improve manuscript:

1-)you can write the title concisely.

2-)the title of the study should give more information regarding the content of the study.

3-)you can add more keywords related to your study.

4-)you should support the statement with the reference. if the reference cannot support you can consider modifying the text.

Functional studies of candidate genes and their locations within genomic regions 62
linked to schizophrenia (SCZ) or bipolar disorder (BD) through linkage studies were prev- 63
alent until around 2006, when genome-disease association studies began to explore po- 64
tential risk gene variants. 

5-)you can give more information about the mechanisms.

More importantly, many genes have been reported to influence the pathogenesis 72
of BD through multiple possible mechanisms [4

6-)you can also add more references to support line 70.

7-)you can give more information about TPM array in the methods section.

8-)you can give more information about Academia Sinica

9-)you can give more information about inclusion exclusion criteria.

10-)you can clearly explain following sentence:

 and highly significant p-values (less than 1 × l0- 112
5) were selected for further analysis. 

11-)you can give more information about how to diagnose patients with BD.

12-)I am not sure whats the P threeshold for the study. the word highly significant sounds unusual to me.

13-)you can improve the table 3.

Author Response

(The authors gave the same response as above.)

Round 2

Reviewer 1 Report

Comments and Suggestions for Authors

No further comments

Reviewer 3 Report

Comments and Suggestions for Authors

The authors improved the manuscript sufficiently.